# Magnetic Evaluation of Heat-Resistant Martensitic Steel Subjected to Microstructure Degradation

**DOI:** 10.3390/ma15144865

**Published:** 2022-07-13

**Authors:** Yi Li, Chao Sun, Kai Liu, Tong Xu, Binbin He

**Affiliations:** 1Department of Mechanical and Energy Engineering, Southern University of Science and Technology, Shenzhen 518055, China; li2002yi2005@163.com; 2China Special Equipment Inspection and Research Institute, Beijing 100029, China; cms12063@126.com (C.S.); lkl20_08@163.com (K.L.)

**Keywords:** microstructure degradation, coercivity, hysteresis loop, heat-resistant steel

## Abstract

The present paper investigates the use of the magnetic hysteresis loop technique to nondestructively evaluate microstructural degradation in heat-resistant martensitic (HRM) steels. The degradation impairs the safe operation of thermal power plants and it is thus essential to periodically assess it using nondestructive evaluation (NDE) techniques. In this contribution, HRM steels are thermally aged up to 16,000 h at 675 °C to simulate the microstructural degradation, then the changes in the magnetic coercivity, hardness, and microstructure are systematically characterized and the relations between them are determined. Both coercivity and hardness decrease with thermal aging duration, which can be interpreted in terms of the microstructure parameters’ evolution based on the pinning of crystal defects on domain walls and dislocations. Coercivity and hardness share the same softening trend with aging time, and good linear relations between coercivity, hardness, and microstructure parameters are found. These results provide a key to understanding the magnetic parameter evolution in HRM steels and suggest the possibility of using magnetic technologies for the NDE of microstructure degradation in thermal power plants.

## 1. Introduction

Heat-resistant martensitic (HRM) steels are widely employed in thermal and nuclear power plants, owing to their good creep and oxidation resistance, together with excellent thermal properties and low cost as compared to bainitic steels and austenitic steels [1,2,3,4,5,6]. The microstructural degradation of HRM steels during operation at elevated temperatures impairs the performance of these steels in thermal and nuclear power plants. This degradation arises from microstructure changes, including the coarsening of precipitates, recovery of dislocations, and recrystallization of martensitic laths [7,8,9,10]. Pipeline segments located at bending elbows, the heat-affected zones of welding joints, and excessively higher-temperature locations are prone to premature failure due to rapid microstructural degradation [11,12,13], which is a key issue concerning the safe operation of power plants. As a consequence, it is essential to periodically assess the microstructural degradation condition of HRM steels using nondestructive evaluation (NDE) techniques.

Currently, the microstructural degradation state is directly investigated using surface metallographic replication or inferred from surface hardness; however, these techniques cannot shed light on the interior of steels. Conventional ultrasonic techniques based on backscatter or velocity are only applicable at the last stage, prior to failure. Thus, alternative NDE techniques capable of revealing the microstructural degradation condition of the interior of HRM steels during their whole service life are desired. Specifically, magnetic hysteresis loop (HL) technology attracts great attention because magnetic properties are sensitive to the evolutions of crystal defects, such as dislocations, precipitates, and boundaries [14,15,16,17], and is thus a promising means for detecting microstructural changes in HRM steels. Moreover, magnetic techniques are able to investigate the microstructure beneath the surface, which can be easily achieved by tuning the frequency of the excitation current.

The changing trend of magnetic coercivity in served, aged, or crept HRM steels has been investigated in previous reports. For instance, Liu et al. [18], Karimian et al. [19], and Wilson et al. [20] investigated the changing trend of magnetic parameters in P9 steel after approximately 11 years’ service at 520 °C and found that the coercivity had decreased. Ryu et al. [21] found that the coercivity of 12Cr-W steels monotonously decreased with thermal aging time at 700 °C, and Bong et al. [22] reported the same tendency in 9Cr-1Mo steels aged at 690 °C. As for the creep test, the coercivity of 11Cr-3.45W steels crept at 700 °C and 60 MPa decreased with duration [23], while that of P91 steels crept at 500 °C and 290 MPa increased with duration [24]. Although these reports demonstrate the potential of using the magnetic technique for the nondestructive evaluation of HRM steels, the quantitative dependance of magnetic parameters on crystal defects still remains unclear, which is the fundamental of NDE of HRM steels and needs to be determined.

In this study, the relationships between coercivity, hardness, and crystal defects are explored. The microstructure parameters, including the spacing of precipitates, dislocation density, and martensitic lath width of the thermally aged P91 steel at different aging durations, are determined based on systematic microscopic analysis. The magnetic and mechanical properties are acquired via HL and hardness measurements, respectively, and their behaviors are interpreted based on the hindrance of crystal defects to the movement of domain walls (DWs) and dislocations. The corresponding relationships between magnetic properties, mechanical properties, and crystal defects are thus investigated based on these analyses.

## 2. Materials and Experiment Methods

The steels investigated in the present study were removed from the commercial P91 pipe (outer diameter: 508 mm; thickness: 62 mm; normalizing temperature: 1047–1067 °C; tempering temperature: 760–773 °C). The chemical composition of the investigated steel is shown in Table 1. To simulate the microstructure degradation of P91 steel during service, accelerated aging at a higher temperature (675 °C) than the service temperature (about 593 °C) was performed by isothermally holding the steels for different durations (50, 100, 200, 500, 1000, 2000, 4000, 8000, and 16,000 h). The Larson–Miller method can be used to connect the aging time to the service condition. By using the Larson–Miller (LM) relationship, (lg*t*_1_ + C)*T*_1_ = (lg*t*_2_ + C)*T*_2_, where *t* is time in hours, *T* is temperature in K, and C is LM constant, (9.7 for P91 steel [5]), the accelerated aging duration of 16,000 h at 675 °C matches about 40 years’ service at 593 °C, which is the average lifetime of power plants. The size of the aged steels was about 80 mm × 62 mm × 20 mm, and the specimens used for magnetic, mechanical, and microstructural evaluation were cut out from these aged steels. Though these steels were aged at 675 °C in air for a long duration, the influence of oxidation could be avoided by cutting the specimens far away (more than 3 mm) from the oxidation surface.

The effects of the evolution of microstructure on the aspects of precipitation size and spacing, dislocation density, and martensitic lath width for the thermally aged P91 steels at varying durations were systematically observed using electron microscopy. The size and spacing of precipitates were characterized using scanning electron microscopy (SEM). The spacing of precipitates was calculated using *s* = (*A*/*N*)^0.5^, where *s* is the spacing of precipitates, *A* is the area of the field of SEM view, and *N* is the number of precipitates in this field. At least 250 precipitates located at 10 different fields of view were employed to calculate the average size and spacing of precipitates. The lath width of martensite and the recovery of dislocations were determined using transmission electron microscopy (TEM). The lath width is defined as the length of the minor axis of the martensitic lath. At least 10 different locations with an area of 70 μm^2^ each were chosen to evaluate the lath width of martensite. The specimens used for TEM observation were prepared through a twin jet electro-polisher (Struers TenuPol-5) in a solution of 5% perchloric acid and 95% ethanol (Vol. %) at −25 ℃. The dislocation density was calculated based on the X-ray diffraction (XRD) using the Williamson–Hall method [25]. The XRD measurement was carried out using Co Kα radiation (~1.788 Å) under a voltage of 30 kV. The precipitations were separated out from the matrix by electrolyzing, and the obtained powder was then analyzed using XRD and Inductively Coupled Plasma-Atomic Emission Spectrometry (ICP-AES) to reveal the crystal structure, weight percentage, and composition of the precipitations. The Vickers hardness was determined from an electro-polished specimen with a size of 10 mm × 10 mm × 5 mm, under a load of 5 kg, with at least ten measurements.

Magnetic HLs were measured using a vibrating sample magnetometer (VSM). Specimens with a length of 6 mm, width of 4 mm, and thickness of 0.6 mm were cut out by electrical discharge machining, ground, and further electro-polished to a thickness of 0.2 mm for VSM measurements. The maximum applied magnetic field was 10,000 Oe (796,000 Am^−1^), with a fine step size of 1 Oe in the regime of −50 and +50 Oe to acquire a more accurate coercivity.

## 3. Results

### 3.1. Decreases in Coercivity and Hardness

The hysteresis loops of P91 steels aged under varied durations are displayed in Figure 1a. Although no obvious difference in maximum magnetization was observed among the present P91 steels with different aging times, the coercivity demonstrated a wide variation, as shown in Figure 1b, which magnified the hysteresis loops around the zero point. Figure 1c shows the coercivity values of P91 steels aged for various times, from which it can be seen that the coercivity sharply decreased at the early aging stage, followed by it reaching a plateau with a nearly constant value. The coercivity decreased monotonously and its reduction ratio surpasses 30%, which indicates that coercivity is sensitive to thermal aging duration and suitable for the NDE of the microstructure evolution of HRM steels.

The Vickers hardness of the aged P91 steels is shown in Figure 2. Similar to the changing trend of coercivity, the hardness sharply decreased from 243, for the as-received steel, to 214 after aging for 500 h, then gradually decreased to 188 when aged for 16,000 h. This tendency coincides well with the previous reports on Grade 91 steel crept at 600 °C for up to 100,000 h [9].

### 3.2. Precipitates Ripening and Martensitic Microstructure Recovery

The overall microstructure of the P91 steels thermally aged at 675 °C for various durations was captured by SEM observation (Figure 3).

Precipitations can be found in the SEM image. The type, composition, and weight percentage of the precipitations are shown in Table 2.

The main type of precipitates is M_23_C_6_, which is dominated by the Cr_23_C_6_. MX precipitates composed of V, Nb, C, and N are sparsely distributed in the matrix by TEM observation and are too small to be detected by SEM in this investigation. Laves Phase was not detected in the entire aging duration. The number density and the spacing of the precipitations were measured using the SEM image and are listed in Table 3. There was an intensive distribution of M_23_C_6_ precipitates with a mean spacing of 0.61 μm in the as-received steels. Most of these precipitates segregated at the prior austenite grain boundaries (PAGBs) and sub-grain boundaries (SGBs) (Figure 3a) due to the fact that boundaries serve as the nucleation sites of the precipitates. The number density of M_23_C_6_ precipitates decreases and the corresponding spacing of the precipitates increases (0.81 μm) when aged for 1000 h (Figure 3b). Additionally, some precipitates on PAGBs grew much faster than the ones on SGBs (Figure 3b), which may be attributed to the lower chemical potential and larger diffusion velocity of PAGBs. After aging for 4000 h and 16,000 h, the number density of M_23_C_6_ further decreased, as shown in Figure 3c,d, and the mean spacing of precipitates increased to 1.02 μm at 4000 h and 1.21 μm at 16,000 h. Some SGBs seemed to be free from decorating precipitates, which may have been dissolved during the aging process [26]. These sub-grains appeared to be larger than the ones still pinned by precipitations, which indicates that the distribution of precipitations may vitally affect the sub-grain size.

Detailed TEM observation was carried out on the present steels to reveal the evolution of the substructures with respect to the aging duration (Figure 4). The as-received steel had a lath martensitic microstructure with a distribution of intensive dislocations (Figure 4a). The lath widths of the aged steels were measured from the TEM images and are listed in Table 3. The lath boundaries were relatively straight, and the mean width of the laths was around 0.35 μm for the as-received steels. Interestingly, some lath segments turned out to be clear and free from dislocations after aging for 1000 h (Figure 4b) due to the pronounced reduction in dislocation density resulting from the annihilation and rearrangement of dislocations. Additionally, some lath boundaries became curved and the lath width grew larger (Figure 4b). The lath width grew even further, with the generation of a few polygonized grains, with the increase in aging time to 4000 h (Figure 4c). The recovery of dislocations became more severe after aging at 675 ℃ for 16,000 h (Figure 4d), with the formation of a large number of equiaxial grains. The lath width grew progressively from 0.35 μm prior to the aging process up to 0.54 μm, 0.66 μm, and 0.76 μm after aging treatments of 1000 h, 4000 h, and 16,000 h, respectively.

The measured dislocation density of the as-received steel was 2.3 × 10^14^ m^−2^ (Figure 5), which is comparable to the lath martensite microstructure reported in the literature [26,27,28,29]. The dislocation density of the present steel drastically decreased in the early aging stage prior to 500 h, then reached a nearly constant value with increased aging duration (Figure 5). This reason for this tendency lies in the fact that dislocations within a lath can either annihilate with each other or move and be absorbed into lath boundaries [26], and a larger number density of dislocations and smaller lath width will greatly accelerate the annihilation owing to the stored strain energy.

The measured martensitic lath width and calculated spacing of M_23_C_6_ precipitates in P91 steels are demonstrated in Figure 6. Interestingly, an almost linear relationship was found between the lath width and the precipitation spacing (Figure 6). M_23_C_6_ precipitates along the PAGBs and SGBs in the as-received steels (Figure 3) could pin the boundaries and drag them from movement, therefore preventing the coarsening of the lath structure. Although the volume fraction remained almost the same during the thermal aging process [30,31], these M_23_C_6_ precipitates still underwent Ostwald ripening, i.e., big precipitates grew larger at the expense of dissolving small precipitates, resulting in a decrease in the number density and an increase in precipitate spacing. These changes reduce the pinning effect of M_23_C_6_ and thus lead to the coarsening of the lath width, resulting in a linear relationship between the lath width and the precipitation spacing (Figure 6).

## 4. Discussion

### 4.1. Dependence of Coercivity on Microstructural Evolution

All crystal defects, including precipitates, dislocations, and lath boundaries, can affect the coercivity of HRM steels. The crucial point is to determine the leading factor at different aging stages. The stress field of dislocations can retard the movement of DWs and increase their coercivity. It is reported that coercivity changes linearly with the square root of dislocation density [32,33]. Unlike dislocations, nonmagnetic precipitates can pin the DWs because they can reduce the DW energy when they are embedded in the DWs. Therefore, an increase in precipitation spacing leads to a smaller coercivity. Coercivity is found to show a linear relation with *Nd*^2^ in Fe-Cu alloys due to the presence of Cu-rich precipitates [14,34], where *N* is the number density and *d* is the diameter of precipitates. Because the spacing of precipitates is in proportion to *Nd*^2^, coercivity will also show a linear relation with the spacing of precipitates. For the martensitic lath boundaries, the misorientation angle is very small (several degrees) [29,35], so it may not exert a great influence on coercivity because it is easy for the DWs to overcome the small exchange energy and anisotropy energy barrier across the lath boundary.

The dependence of coercivity on the square root of dislocation density of the aged steels prior to 500 h is depicted in Figure 7a; interestingly, a nearly linear relationship can be observed. Similarly, a perfect linear relation was also found among the measured coercivity and spacing of M_23_C_6_ precipitates of the present P91 steels with aging durations larger than 500 h (Figure 7b). Therefore, it can be inferred from the changing tendency of coercivity, dislocations, and precipitates that the sharp reduction in coercivity at the early aging stage before 500 h mainly stems from the fast recovery of dislocations. Sternberk et. al. [33] also found that in short-time-tempered low alloy Cr-Mo steels the decisive factors which control the coercive force are the dislocations rather than the precipitates. The dislocation density for the late aging stage beyond 500 h remained nearly constant at low values (Figure 5), while the spacing of precipitates increased from 0.7 μm to 1.2 μm (Figure 3); consequently, the precipitates are believed to play an important role in controlling the coercivity at the late aging stage.

Our results show that not only precipitates but also dislocations play an important role in affecting the coercivity in thermally aged P91 steels, while some results achieved on heat-resistant bainitic steels (2.25Cr-1Mo) mainly ascribe the changes in coercivity to the evolution of precipitates [36,37]. This discrepancy may arise from the different initial dislocation density. It is known that lath martensite possesses a much larger number density of dislocations as compared with bainite microstructures (2.25Cr-1Mo steel). The present finding can also explain the increasing trend of coercivity in the literature [23,24], which lies in the multiplication of dislocation due to the excessively large applied stress in the literature.

### 4.2. Relation between Mechanical Properties and Coercivity

The relation between hardness and coercivity is shown in Figure 8, from which a two-stage linear relationship is discovered as follows:(1)$$HV=7.8H_{c}+148\;(t\;<\;500\;h)$$
(2)$$HV=35H_{c}-90\;(t\;>\;500\;h)$$
where *HV* is Vickers hardness, *H_c_* is coercivity in Oe, and *t* is the aging time. The dislocations play the leading role in affecting the hardness and coercivity in the early stage (*t* < 500 h). In particular, the hardness also varies with the square root of dislocations following Taylor’s law [38]:(3)$$\sigma_{\rho }=0.5MGb\sqrt{\rho }$$
where *σ_ρ_* is dislocation hardening, *M* is the Taylor factor, *G* is the shear modulus, *b* is Burgers vector, and *ρ* is dislocation density. Therefore, the relationship between hardness and coercivity is linear. With the increase in aging time beyond 500 h, the precipitates play an important role in governing the hardness and coercivity during prolonged aging larger than 500 h. The coercivity linearly varies with the reciprocal of precipitation spacing (Figure 7b). The strength induced by precipitations is also in proportion to the reciprocal of precipitation spacing according to Orowan’s equation [39]:(4)$$\sigma_{p}=MGb/\lambda$$
where *σ*_p_ is the precipitation hardening and *λ* is the mean spacing of the precipitates. Therefore, a linear relationship shall exist between hardness and coercivity, which is confirmed by Equation (2). The present two-stage linear relationship between coercivity and hardness is different from the single linear relationship reported on the heat-resistant bainitic steels [36,40], which may be attributed to the larger initial number density of dislocations in martensite compared to bainite microstructures. The linear relation between coercivity and hardness, as shown in the present investigation, strongly suggests that it is desirable to use the magnetic technique for the NDE of thermal degradation of P91 steels in power plants.

## 5. Conclusions

In this study, the evolutions of the magnetic properties, mechanical properties, and microstructure parameters of P91 HRM steels after thermal aging were investigated, and the relations between them were determined. The coercivity and hardness decreased with thermal aging duration due to dislocation recovery, the coarsening of M_23_C_6_ precipitates, and the growth of martensitic laths. The recovery of dislocations plays a principal role in affecting the coercivity at the early aging stage, and the ripening of M_23_C_6_ precipitates exerts an important influence on coercivity at the late aging stage. Coercivity and hardness share the same softening trend with aging time, and a good two-stage linear connection between coercivity and hardness was found. These results provide a key to understanding the magnetic parameter evolution in HRM steels and suggest the possibility of using magnetic technologies for the NDE of microstructure degradation in thermal power plants.

## Figures and Tables

**Figure 1 materials-15-04865-f001:**
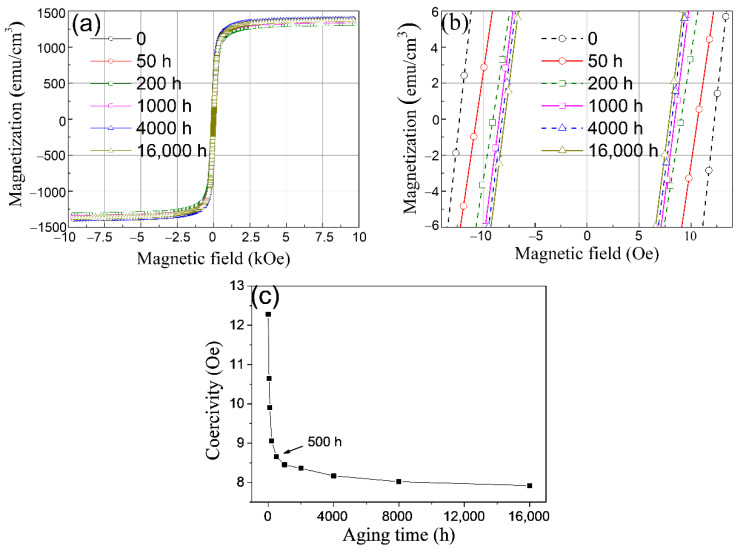
(**a**) Hysteresis loops of P91 steels processed by aging at 675 °C for different durations. (**b**) The magnification of the hysteresis loops around the zero point. (**c**) Coercivity of P91 steels thermally aged for various times. Coercivity is the magnetic field when the magnetization is zero.

**Figure 2 materials-15-04865-f002:**
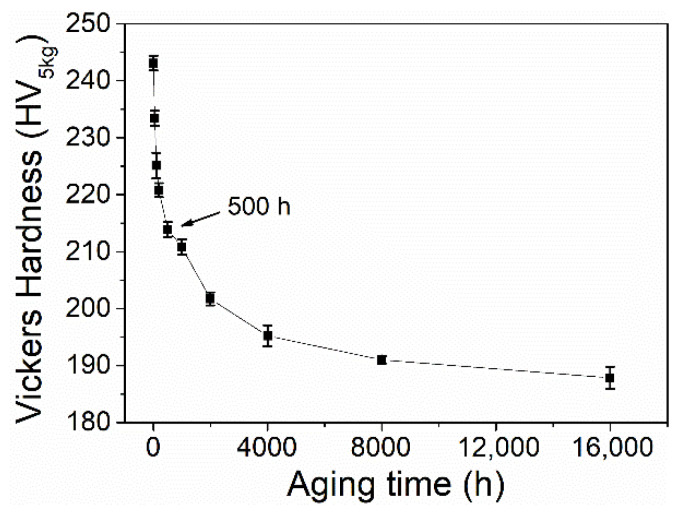
Vickers hardness of the present steels with varying aging times. The error bar represents the 95% confidence interval of the measurements.

**Figure 3 materials-15-04865-f003:**
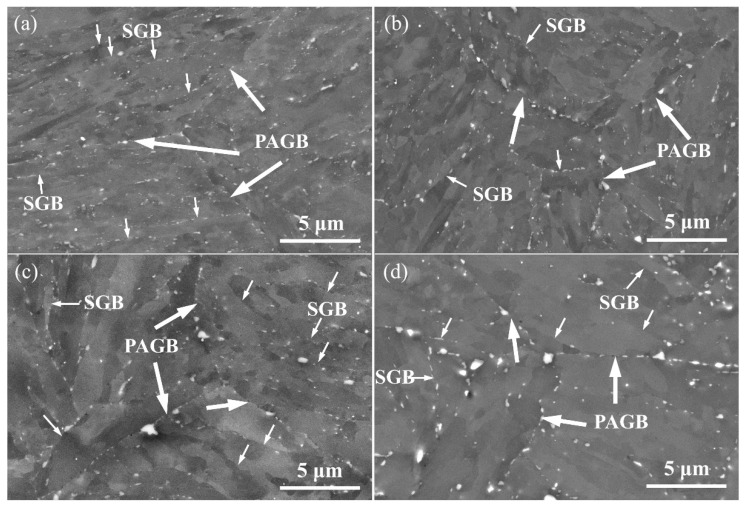
SEM images illustrating the coarsening of M_23_C_6_ precipitates with the increase in duration at 675 °C for P91 steels, including (**a**) 0 h, (**b**) 1000 h, (**c**) 4000 h, and (**d**) 16,000 h. The bright particles are M_23_C_6_ precipitates. The wide arrows mark the prior austenite grain boundaries (PAGBs) and the narrow arrows denote the sub-grain boundaries (SGBs). Most M_23_C_6_ precipitates are located at these boundaries.

**Figure 4 materials-15-04865-f004:**
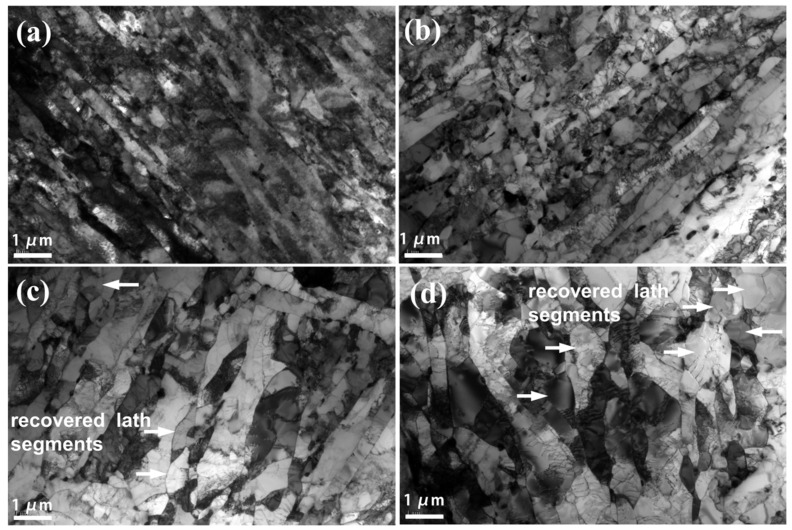
TEM images showing the recovery of martensitic lath structure in the present Gr.91 steels with different durations at 675 °C, including (**a**) 0 h, (**b**) 1000 h, (**c**) 4000 h, and (**d**) 16,000 h. The arrows in (**c**,**d**) indicate some recovered lath segments.

**Figure 5 materials-15-04865-f005:**
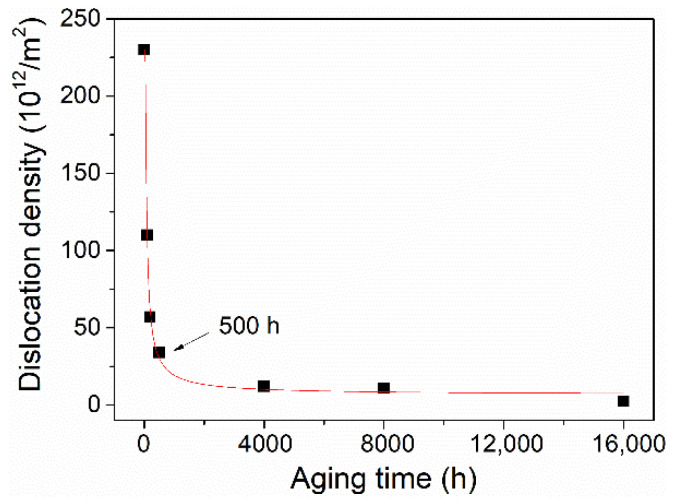
The evolution of dislocation density in the present P91 steels with respect to their different aging durations.

**Figure 6 materials-15-04865-f006:**
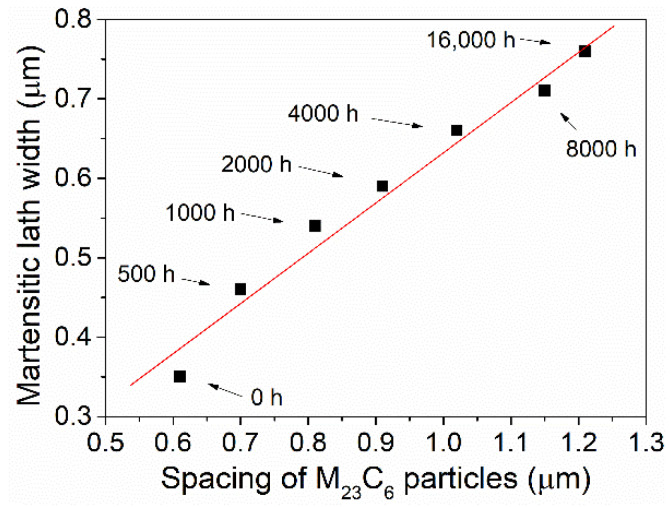
The relation between the lath width of martensite and the spacing of M_23_C_6_ precipitates.

**Figure 7 materials-15-04865-f007:**
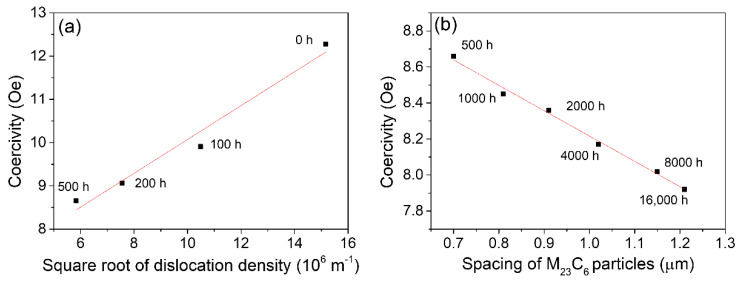
(**a**) Dependence of coercivity on the square root of dislocation density during the early aging stage (<500 h); (**b**) dependence of coercivity on the spacing of M_23_C_6_ precipitates at the aging stage beyond 500 h.

**Figure 8 materials-15-04865-f008:**
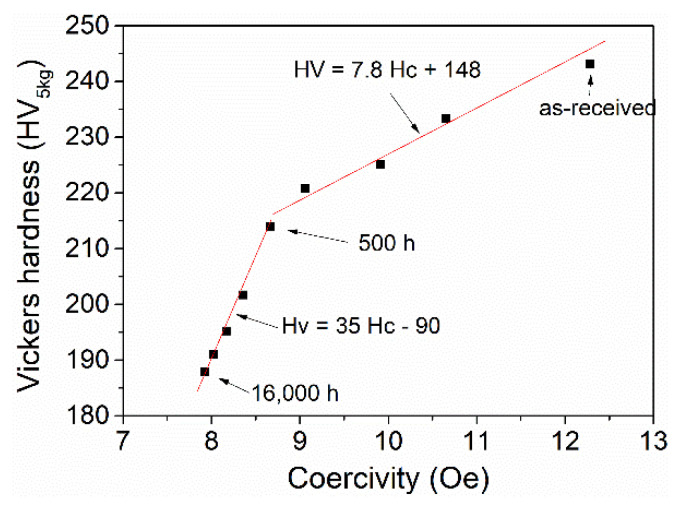
The relation between Vickers hardness and coercivity in the present steel with different age durations.

**Table 1 materials-15-04865-t001:** Chemical composition of the investigated steel (weight percentage, wt%).

C	Si	Mn	P	S	Cr	Mo
0.10	0.34	0.50	0.012	0.005	8.73	0.94
Al	V	Nb	Ti	N	Ni	Cu
0.01	0.216	0.073	0.002	0.0384	0.06	0.04

**Table 2 materials-15-04865-t002:** Type, composition, and weight percentage (wt.%) of the precipitations in aged P91 steels.

Aging Time (h)	Type	Composition	wt.%
0	M_23_C_6_	(Cr_0.662_Fe_0.265_Mo_0.054_Mn_0.011_V_0.007_Ni_0.001_)_23_C_6_	1.49
	MX	(Nb_0.188_V_0.490_Cr_0.288_Mo_0.027_Ti_0.007_)(C_0.335_N_0.665_)	0.29
4000	M_23_C_6_	(Cr_0.717_Fe_0.206_Mo_0.061_Mn_0.010_V_0.005_Ni_0.001_)_23_C_6_	1.57
	MX	(Nb_0.190_V_0.524_Cr_0.256_Mo_0.022_Ti_0.008_)(C_0.363_N_0.637_)	0.30
16,000	M_23_C_6_	(Cr_0.713_Fe_0.210_Mo_0.061_Mn_0.010_V_0.004_Ni_0.002_)_23_C_6_	1.61
	MX	(Nb_0.175_V_0.506_Cr_0.276_Mo_0.033_Ti_0.010_)(C_0.319_N_0.681_)	0.30

**Table 3 materials-15-04865-t003:** Number density and spacing of the precipitations and martensitic lath width of the aged samples.

Aging Duration (h)	Number Density (10^12^/m^2^) Mean ± 95%CI	Carbide Spacing (μm) Mean ± 95%CI	Lath Width (μm) Mean ± 95%CI
0	2.69 ± 0.46	0.61 ± 0.06	0.35 ± 0.07
500	2.04 ± 0.35	0.70 ± 0.07	0.46 ± 0.09
1000	1.52 ± 0.29	0.81 ± 0.09	0.54 ± 0.11
2000	1.21 ± 0.28	0.91 ± 0.13	0.59 ± 0.1
4000	0.96 ± 0.21	1.02 ± 0.13	0.66 ± 0.12
8000	0.76 ± 0.17	1.15 ± 0.16	0.71 ± 0.11
16,000	0.68 ± 0.16	1.21 ± 0.17	0.76 ± 0.11

## Data Availability

All data included in this study are available upon request by contact with the corresponding author.

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
