# Peer review of "Magnetic Evaluation of Heat-Resistant Martensitic Steel Subjected to Microstructure Degradation"

_materials, 2022, doi:10.3390/ma15144865_

Round 1

Reviewer 1 Report

The authors studied the evolutions of the magnetic properties, mechanical propertie, and microstructure of P91 HRM steel after long-time thermal aging. The experiments were conducted in a very good standard and the paper is well written. The results are convincing and the magnetic technology seems a good method for the NDE of microstructure degradation in thermal power plants. Therefore, the paper is recommended for publication without any changes. Just one question for the authors to consider.

-The ageing temperature (675C) is already higher than the normal service temperature (lower than 600C) to accelerate the experiments. Then why the duration of 16000 hours is still necessary? This test takes almost 2 years. Any good reason for that?

Author Response

Reviewer 1: The authors studied the evolutions of the magnetic properties, mechanical properties, and microstructure of P91 HRM steel after long-time thermal aging. The experiments were conducted in a very good standard and the paper is well written. The results are convincing and the magnetic technology seems a good method for the NDE of microstructure degradation in thermal power plants. Therefore, the paper is recommended for publication without any changes. Just one question for the authors to consider.

Comment: - The ageing temperature (675C) is already higher than the normal service temperature (lower than 600C) to accelerate the experiments. Then why the duration of 16000 hours is still necessary? This test takes almost 2 years. Any good reason for that?

Reply: Thank you for the comment! The normal service temperature of P91 HRM is about 593 oC, and the average lifetime of power plants is about 40 years. By using the Larson-Miller (LM) relationship , where t is time in hours, T is temperature in K, C is LM constant (9.7 for P91 steel), the accelerated aging duration of 16000 hours at 675 °C matches about 40 years’ service at 593°C, which is the average lifetime of power plants.

We have added some sentences to clarify this problem. Please refer to Line 79-83 in the revised manuscript where the corrections are marked in red.

Reviewer 2 Report

Present paper is focused on the implementation of nondestructive magnetic measurements for evaluation of martensitic steel microstructural degradation. Interesting relations between various microstructural and mechanical parameters are experimentally measured and demonstrated. The findings of the work can be interesting both within academical and industrial areas. In general, the work gives a positive impression, however, there are several remarks, which are hoped to be addressed:

Line 62: “The lath thickness is defined as the length of the minor axis if the lath structure partially polygonises”

This sentence meaning is unclear, please, consider rewriting it.

Line 139: “The main type of precipitates is M23C6”

How exactly did the authors have identified the carbide type and that it is a main one? Could they provide an experimental verification of this statement?

Please, provide not only the mean values but also standard deviations for carbide-related (spacing, density, etc.) and grain-related (lath thickness, etc.) measured values within all four observed conditions in a single Table. It would be great to see all the other measurements (dislocation density, coercivity, hardness, etc.) also in the same Table.

Please be careful, 'correlation' commonly presumes certain computations of correlation coefficients etc. In other cases, it is better to use 'relations'.

Despite they are well known, Taylor’s and Orowan’s laws should be written on a separate line as Equations (3) and (4).

The curves of coercivity (Fig 1c) and dislocation density (Fig 5) vs aging time have similar shape and corresponding measurement points match perfectly with each other. Thus, one can expect linear relation between these two parameters. However, Fig 7 shows coercivity vs square root of dislocation density instead of just dislocation density. It looks like that a second order fit line on Fig 7 would fit much better than a linear one, which again indicates linear relation between coercivity and dislocation density. Also, the graph on Fig 7 is shown not within a whole range (0 – 16 000 h) but only from 0 to 500 h – why not to show the whole range? If the authors state that dislocations play role only up to 500 h, then one could see that on the whole range as well.

Author Response

Reviewer 2: Present paper is focused on the implementation of nondestructive magnetic measurements for evaluation of martensitic steel microstructural degradation. Interesting relations between various microstructural and mechanical parameters are experimentally measured and demonstrated. The findings of the work can be interesting both within academical and industrial areas. In general, the work gives a positive impression, however, there are several remarks, which are hoped to be addressed:

Comment: - Line 62: “The lath thickness is defined as the length of the minor axis if the lath structure partially polygonises”. This sentence meaning is unclear, please, consider rewriting it.

Reply: Thank you for the comment! In fact, “lath width” or “lath size” is more commonly used in literature. Here we change this sentence to “the lath width is defined as the length of the minor axis of the martensitic lath”, which is obviously applicable for both the original and the recovery state.

We have rewritten the sentence. Please refer to Line 96-97 in the revised manuscript where the corrections are marked in red.

Comment: - Line 139: “The main type of precipitates is M23C6”, How exactly did the authors have identified the carbide type and that it is a main one? Could they provide an experimental verification of this statement?

Reply: Thank you for the comment! The precipitations are separated out from the matrix by electrolyzing, and the obtained powder is then analyzed by XRD and Inductively Coupled Plasma-Atomic Emission Spectrometry (ICP-AES) to reveal the crystal structure, weight percentage, and composition of the precipitations. The corresponding results are added and shown in Table 2.

  We added the precipitation identification results in Table 2 in the revised manuscript, and the corresponding experiment methods are added in line 103-106. Please refer to it.

Comment: - Please, provide not only the mean values but also standard deviations for carbide-related (spacing, density, etc.) and grain-related (lath thickness, etc.) measured values within all four observed conditions in a single Table. It would be great to see all the other measurements (dislocation density, coercivity, hardness, etc.) also in the same Table.

Reply: Thank you for the suggestions! We have added Table 3 to show the measured data with error bar following your suggestions.

Please refer to Table 3 in the revised manuscript.

Comment: - Please be careful, 'correlation' commonly presumes certain computations of correlation coefficients etc. In other cases, it is better to use 'relations'.

Reply: Thank you for pointing out this! We have changed 'correlation' to 'relation' in the revised manuscript.

Comment: - Despite they are well known, Taylor’s and Orowan’s laws should be written on a separate line as Equations (3) and (4).

Reply: Thank you for the suggestion! We have written Equations (3) and (4) on a separate line following your suggestion. Please refer to equation (3) and (4) in the revised manuscript.

Comment: - The curves of coercivity (Fig 1c) and dislocation density (Fig 5) vs aging time have similar shape and corresponding measurement points match perfectly with each other. Thus, one can expect linear relation between these two parameters. However, Fig 7 shows coercivity vs square root of dislocation density instead of just dislocation density. It looks like that a second order fit line on Fig 7 would fit much better than a linear one, which again indicates linear relation between coercivity and dislocation density. Also, the graph on Fig 7 is shown not within a whole range (0 – 16 000 h) but only from 0 to 500 h – why not to show the whole range? If the authors state that dislocations play role only up to 500 h, then one could see that on the whole range as well.

Reply: Thank you for the comment!

In fact, either coercivity vs dislocation density or coercivity vs square root of dislocation density shows good linear relation (<500 h) according to our results. Since it is widely proved in literature that the coercivity changes linearly with the square root of dislocation density, we incline to plot coercivity vs square root of dislocation density in this study.

Beyond 500 h, the above linear relation does not fit well. In this aging stage, the dislocation density seems to be invariable, but the coercivity decreases continuously, which may be mainly due to the influence of precipitation ripening. So we conclude that the coercivity is mainly affected by dislocations in the aging stage within 500 h.

Reviewer 3 Report

Manuscript ID: materials-1789130

Title: Magnetic evaluation of heat-resistant martensitic steel subjected to
microstructure degradation

Journal: Materials (MDPI)

In this work, study the foundation of using magnetic hysteresis loop technique to non-destructively evaluate the microstructural degradation in heat-resistant martensitic (HRM) steels. HRM steels are thermally aged up to 16,000 hours at 675 °C to simulate the microstructural degradation, then the changes of the magnetic coercivity, hardness, and microstructure are systematically characterized and reported. Further, the correlations between them are created. The obtained results are interesting and shows the reasonable novelty and useful to thermal power industries. Therefore, I recommend to accept the article in Journal “Materials” after clarifying the below given points.

Major revision:

·         It is necessary to clearly show in the introduction what is the novelty and originality of this work.

·         Introduction section need to be strengthen with importance of martensitic steel and ageing requirements while comparing with existing one, application with proper literature. Proper research gap need to be explained in details.

·         How authors were identified and classified the different hours ageing duration. Do you follow any scientific background to fix the ageing duration? Justify clearly in the manuscript.

·         If possible, try to include one higher magnification TEM image for lower ageing duration and higher ageing duration to show the clear difference.

·         The hardness of the lower aging time (500 hr) is quite large (215 HV5Kg) while compared with higher aging time. It is necessary to analyze the reasons for this in depth.

·         If possible, try to provide the schematic diagram for dislocation density vs aging durations for more and easy understanding to common readers.

·         Keep the same unit/notations for the microhardness.

·         Authors are recommended to provide the original origin figures. Present figures show with lower resolution.  

·         Authors are recommended to provide the chemical composition graph, atleast for lower and higher aging durations.

·         Figure 8 is mismatching in running text while compared with figure caption. Check and correct it.

·         Figure.3 should be described in more detail in the text

Author Response

Reviewer 3: In this work, study the foundation of using magnetic hysteresis loop technique to non-destructively evaluate the microstructural degradation in heat-resistant martensitic (HRM) steels. HRM steels are thermally aged up to 16,000 hours at 675 °C to simulate the microstructural degradation, then the changes of the magnetic coercivity, hardness, and microstructure are systematically characterized and reported. Further, the correlations between them are created. The obtained results are interesting and shows the reasonable novelty and useful to thermal power industries. Therefore, I recommend to accept the article in Journal “Materials” after clarifying the below given points.

Comment: - It is necessary to clearly show in the introduction what is the novelty and originality of this work.

Reply: Thank you for the comment! We reveal the quantitative dependance of coercivity on crystal defects in the degraded HRM steels, which is the fundamental of NDE of HRM steels but remains unclear in the former studies. We present it in the last sentences of paragraph 2 in the introduction in the revised manuscript. Please refer to line 59-61 in the revised manuscript.

Comment: - Introduction section need to be strengthen with importance of martensitic steel and ageing requirements while comparing with existing one, application with proper literature. Proper research gap need to be explained in details.

Reply: Thank you for the suggestion! We added some sentences to strengthen the introduction following your suggestion. Please refer to line 1-3 in the revised manuscript.

Comment: - How authors were identified and classified the different hours ageing duration. Do you follow any scientific background to fix the ageing duration? Justify clearly in the manuscript.

Reply: Thank you for the comment!

The normal service temperature of P91 HRM is about 593 oC, and the average lifetime of power plants is about 40 years. By using the Larson-Miller (LM) relationship , where t is time in hours, T is temperature in K, C is LM constant (9.7 for P91 steel), the accelerated aging duration of 16000 hours at 675 °C matches about 40 years’ service at 593°C, which is the average lifetime of power plants.

   We have added some sentence to clarify the problem following your suggestion. Please refer to line 79-83 in the revised manuscript.

Comment: - If possible, try to include one higher magnification TEM image for lower ageing duration and higher ageing duration to show the clear difference.

Reply: Thank you for the suggestion! It is regrettable that the suitable higher magnification TEM image is not accessible right now. Though the details of higher magnification are not clear enough, the current magnification images are capable to show the differences of martensitic lath width and dislocation density. We will deliver high resolution images to the editor, then the images would be much clearer if published.

Comment: - The hardness of the lower aging time (500 hr) is quite large (215 HV5Kg) while compared with higher aging time. It is necessary to analyze the reasons for this in depth.

Reply: Thank you for the comment! The hardness is affected by many factors including: dislocation density, precipitations, sub-grain size. Although the dislocation density remains relatively stable beyond the aging time of 500 hours, the increasement of lath width and the spacing of precipitations are still large, which could further decrease the hardness. However, the data including yield strength, ultimate tensile strength, the misorientation of martensitic lath and block boundaries, and the martensitic block size is needed to quantitatively analyze the changes of hardness, which is our work in the near future.

Comment: - If possible, try to provide the schematic diagram for dislocation density vs aging durations for more and easy understanding to common readers.

Reply: Thank you for the suggestion! The schematic diagram for dislocation density vs aging durations is shown in Figure 5.

Comment: - Keep the same unit/notations for the microhardness.

Reply: Thank you for the comment! We have unified the unit/notations for the microhardness following your comment, please refer to figure 2 and 8 in the revised manuscript.

Comment: - Authors are recommended to provide the original origin figures. Present figures show with lower resolution.  

Reply: Thank you for the comment! We will provide the original origin figures of high resolution to the editor, then the figures would be much clearer if published.

Comment: - Authors are recommended to provide the chemical composition graph, at least for lower and higher aging durations.

Reply: Thank you for the suggestion!

The chemical composition of the initial steel is listed in Table 1. Aging processing would not change the chemical composition of the steel.

The compositions of the precipitations of the initial, 4000 h, and 16000 h sample are added in Table 2 in the revised manuscript. Please refer to it.

Comment: - Figure 8 is mismatching in running text while compared with figure caption. Check and correct it.

Reply: Thank you for pointing out this mistake! We have made the corrections in the revised manuscript. Please refer to line 287 in the revised manuscript.

Comment: - Figure.3 should be described in more detail in the text

Reply: Thank you for the comment! We have added table 2, table 3, and some sentences (line 142-144, 147-148, 155-156, 160-162) to describe figure 3. Please refer to these changes in the revised manuscript.